# Combined Application of Coffee Husk Compost and Inorganic Fertilizer to Improve the Soil Ecological Environment and Photosynthetic Characteristics of Arabica Coffee

**Zeyin Jiang** [1,2,3,†], **Yuqiang Lou** [4,†], **Xiaogang Liu** [1,2,3,*], **Wenyan Sun** [1,2,3], **Haidong Wang** [1,2,3], **Jiaping Liang** [1,2,3], **Jinjin Guo** [1,2,3], **Na Li** [1,2,3] and **Qiliang Yang** [1,2,3]

1   Faculty of Modern Agricultural Engineering, Kunming University of Science and Technology, Kunming 650500, China; jiangzeyin7844@163.com (Z.J.); 18298387516@163.com (W.S.); wanghd@nwsuaf.edu.cn (H.W.); 20220011@kust.edu.cn (N.L.)
2   Yunnan Provincial Field Scientific Observation and Research Station on Water-Soil-Crop System in Seasonal Arid Region, Kunming University of Science and Technology, Kunming 650500, China
3   Yunnan Provincial Key Laboratory of High-Efficiency Water Use and Green Production of Characteristic Crops in Universities, Kunming University of Science and Technology, Kunming 650500, China
4   Institute of Tropical and Subtropical Cash Crops, Yunnan Academy of Agricultural Sciences, Baoshan 678000, China; louyuqiang@126.com
*   Correspondence: liuxiaogangjy@126.com
†   These authors contributed equally to this work.

**Abstract:** Excessive use of chemical fertilizers deteriorates the soil environment and limits the normal growth of Arabica coffee trees. In order to identify the optimal coupling mode of chemical fertilizer application and biomass return that enhances the soil ecological environment and promotes the photosynthetic efficiency of Arabica coffee, this study investigated the impacts of three levels of inorganic fertilizers ($F_L$: 360 kg·ha$^{-1}$, $F_M$: 720 kg·ha$^{-1}$, and $F_H$: 1080 kg·ha$^{-1}$) and three types of coffee husk returning methods ($C_B$: coffee husk biochar, $C_C$: coffee husk compost, $C_A$: coffee husk ash) on the soil fertility, microbial amount, enzyme activity, and photosynthetic characteristics of the Arabica coffee root zone. The entropy weight-TOPSIS method was employed to evaluate the comprehensive benefits. The results showed that $F_M$ had the biggest effect on improving soil fertility, microorganisms, and enzyme activities compared with $F_L$ and $F_H$. Moreover, compared to $C_A$, $C_C$ significantly increased soil organic carbon, organic matter, and total nitrogen content. $C_C$ significantly enhanced the activities of soil phosphatase and urease, respectively, by 29.84% and 96.00%, and significantly increased the amount of bacteria, fungi, and actinomycetes by 62.15%, 68.42%, and 46.21%, respectively. The net photosynthetic rate (Pn), transpiration rate (Tr), and stomatal conductance (Gs) of $F_M C_C$ were significantly higher than those of other treatments. The comprehensive benefit evaluation of the soil environment and photosynthetic characteristics by the entropy weight-TOPSIS method ranked $F_M C_C$ first. Therefore, $F_M C_C$ was the optimal coupling mode for fertilizer application and the coffee husk returning method. The findings of this study not only provide scientific guidance for fertilizing Arabica coffee but also clarify the proper approach to returning coffee husk to the field, thereby improving soil ecology and promoting green and efficient production of specialty crops.

**Keywords:** arabica coffee; coffee husk; combined application of organic and inorganic fertilizers; soil ecological environment; comprehensive evaluation

## 1. Introduction

Yunnan Province is the primary producing region of Arabica coffee in China, with its planting area and output value accounting for over 97% of the country [1]. However, excessive application of chemical fertilizers in pursuit of maximum yield is a common practice in coffee cultivation. Additionally, approximately $2 \times 10^5$ tons of coffee husk

were produced every year, and the direct return of untreated coffee husk to the field led to a decline in soil quality [2,3]. Therefore, developing a scientific fertilization system and exploring effective ways to utilize coffee husk at the same time are of great significance for promoting green planting and ensuring the sustainable development of Arabica coffee.

The inappropriate use of chemical fertilizers resulted in various problems, including low fertilizer use efficiency, decreased crop yield and quality, soil acidification, nutrient imbalance, and decreased organic matter content [4,5]. Moreover, it negatively impacted crop root growth and nutrient absorption, disrupted the balance of soil microbial communities, decreased enzyme activity, and hindered crop growth and development [6–8]. On the other hand, proper fertilization practices improved the physicochemical properties of soil in the crop root zone, increased the activities of rhizosphere microorganisms and enzymes, and promoted crop growth [9]. Therefore, finding the appropriate amount of fertilizer is crucial for maintaining a healthy soil environment and ensuring normal crop growth.

Currently, the main approach to utilizing agricultural and forestry waste resources is composting by aerobic fermentation and biochar production by pyrolysis and carbonization [10,11]. Composting followed by returning to the field enhanced soil physicochemical properties, increased soil enzyme activity and microbial biomass, and promoted plant growth [12–14]. Application of biochar produced by pyrolysis might lead to significant improvements in the soil, such as enhanced soil aggregate structure, increased organic matter accumulation, boosted availability of nitrogen and phosphorus [15,16], promoted plant photosynthesis, and improved stomatal conductance and transpiration rate [11,17]. Regarding the utilization of coffee by-products, current research mainly focuses on biochar production and composting coffee grounds. It has been shown that replacing chemical fertilizers with coffee grounds can increase soil respiration rate, soil organic carbon, total nitrogen, and trace element content [18–20], promote nutrient absorption by crops, and boost crop photosynthetic efficiency and yield [21]. Coffee husks, including pectin, are rich in organic and mineral nutrients. Returning coffee husk to the field supplemented soil nutrients, ensured soil nutrient balance, promoted sustainable land use, and realized the recycling of biological resources [22]. However, few studies have explored the effects of different methods of returning coffee husk to the soil environment in the root zone and the physiological state of coffee.

The combined application of organic and inorganic fertilizers can reduce the reliance on chemical fertilizers while increasing soil nutrient content and the amount of bacteria, fungi, and actinomycetes. Furthermore, the physiological state of plants can be improved as well [23]. By substituting 30% of chemical fertilizers with organic fertilizers, the soil bulk density could be reduced while increasing soil organic carbon, organic matter, total nitrogen, and the proportion of medium-large aggregates. This promoted microbial reproduction, boosted enzyme activity such as urease and catalase, and increased leaf area index, photosynthetic rate, and water use efficiency [11,24]. Replacing 50% of nitrogen fertilizer with organic manure improved the soil carbon-to-nitrogen ratio and enzyme activity, which in turn increased chlorophyll content [25]. Research shows that using compost from medicinal plants on marigold improves the productive and qualitative traits of plants [26]. The combined use of biochar, poultry manure compost, and pyroligneous solution could enhance the activities of dehydrogenase, β-glucosidase, and urease enzymes and increase the contents of dissolved organic carbon and microbial biomass carbon [7]. Fertilizer combined with biochar can increase soil ventilation and microbial activity, promote nitrogen and phosphorus cycling, and enhance aggregate stability [17,27]. Furthermore, the use of chemical fertilizers with plant ash can significantly increase the content of total phosphorus and total potassium in the soil while inhibiting the spread of pathogenic microorganisms and improving soil quality [12,28]. Despite the effectiveness of these methods, it is still unclear how to regulate the physicochemical properties of soil, the amount of microorganisms, the enzyme activity, and the photosynthetic characteristics of coffee trees after the combined application of chemical fertilizers with bio-organic fertilizer, biochar, and ash produced from coffee husk. Therefore, further exploration is warranted in this area.

Returning agricultural and forestry waste in a proper manner can significantly enhance the quality of cultivated land and improve crop growth characteristics. However, the optimal method of effectively combining coffee husk with chemical fertilizers to enhance the soil, water, and fertilizer environment and promote coffee tree growth remains elusive, necessitating further research. It is hypothesized that the application of proper methods for returning coffee waste coupled with optimal levels of inorganic fertilizers can comprehensively improve the soil's ecological environment and coffee growth. Therefore, the objective of this study was to investigate the impact of coffee husk returning methods on soil fertility, microorganisms, enzyme activity, and photosynthetic characteristics of Arabica coffee given various levels of inorganic fertilizers based on combined applications of organic and inorganic fertilizers. The ultimate goal was to identify the optimal organic–inorganic fertilizer coupling mode and provide a scientific basis for efficient and high-quality production of Arabica coffee and the sustainable development of the soil environment in Yunnan Province, China.

## 2. Material and Methods

### 2.1. Study Site

The experiment was carried out from March 2020 to September 2021 in the plastic greenhouse of the Faculty of Modern Agricultural Engineering, Kunming University of Science and Technology, Yunnan Province, Southwest China (24°09′ N, 102°79′ E, 1978.9 m a.s.l.). The annual average temperature of the experimental greenhouse was 25 °C, with a relative humidity of 45–70%. The soil pH value was 6.5–7.5, and the soil organic matter content was 15.05 g $kg^{-1}$. Total nitrogen, total phosphorus, and total potassium content were 0.87, 0.68, and 13.90 g $kg^{-1}$, respectively. Nitrate nitrogen, available phosphorus, and available potassium content were 27.48, 112.61, and 85.53 mg $kg^{-1}$, respectively. The experimental crop was 1-year-old young coffee trees with uniform growth.

### 2.2. Experimental Method

The experimental setting was set up for three inorganic fertilizer levels and three kinds of coffee husk returning methods, a complete combination design, a total of nine treatments, each treatment having three replicates, for a total of 27 communities. The experimental setting was set up for three inorganic fertilizer levels and three kinds of coffee husk returning methods, a complete combination design, a total of nine treatments, each treatment having three replicates, for a total of 27 communities. Three coffee husk returning methods were applied: biochar ($C_B$), compost ($C_C$), and ash ($C_A$) under three fertilization levels (low fertilizer ($F_L$): 360 kg $ha^{-1}$, middle fertilizer ($F_M$): 720 kg $ha^{-1}$, and high fertilizer ($F_H$): 1080 kg $ha^{-1}$). Arabica coffee trees (Catimor P7963) with planting spacing and row spacing of 0.8 m and 1.2 m.

Fertilizer used in the experiment was a large-element water-soluble fertilizer ($N:P_2O_5:K_2O$ = 20%:20%:20%, Saigute Biotechnology Co., Ltd., Wuhan, China). Fertilizer was fertigated to the coffee field through drip irrigation systems using a pressure differential tank on 5 March 2020, 9 July 2020, 9 March 2021, and 13 July 2021, respectively. The preparation method and mineral content of coffee husk returning methods were as follows: (1) $C_B$: Fresh coffee husk was dried thoroughly and then pyrolyzed and carbonized at 400 °C; the contents of total nitrogen, total phosphorus, and total potassium were 7.59, 1.92, and 3.22 g·$kg^{-1}$, respectively. (2) $C_C$: The fresh coffee husk was placed in the fermentation tank for 5 weeks of mechanical forced oxygen fermentation, and it was light brown after full rot, and the moisture content was adjusted by 20% for later use; the contents of total nitrogen, total phosphorus, and total potassium were 17.5, 5.31, and 4.36 g·$kg^{-1}$, respectively. (3) $C_A$: After the fresh coffee was fully dried, it was burned fully in an incinerator placed outdoors; the contents of total nitrogen, total phosphorus, and total potassium were 1.03, 1.21, and 3.45 g·$kg^{-1}$, respectively. The application rate was 20 t $ha^{-1}$ fresh coffee husk with a moisture content of 82.5%, regardless of the conversion loss under different treatments. A coffee husk from three returning methods was evenly embedded in

an annular ditch with a depth of 20 cm at a distance of 20 cm from the trunk on 2 March 2020 and 10 March 2021.

A standard evaporating dish with a diameter of 20 cm was set in the center of the greenhouse to measure the water surface evaporation. During the experiment, the coffee trees were irrigated every 7 days with the same irrigation amount, and the irrigation amount was calculated as follows:

$$I_r = K_c \times E_p \times S \tag{1}$$

where $I_r$ is the amount of irrigation per plant (mm), $K_c$ is the crop coefficient ($K_c = 0.95$), $E_p$ is the water surface evaporation (mm) during two irrigation intervals, and $S$ is the irrigation control area (cm$^2$). The irrigation control area in this study was a circle with a radius of 25 cm.

### 2.3. Sampling and Measurements

### 2.3.1. Soil Sample Collection and Analysis

Soil samples of 0–40 cm were collected every 10 cm on 3 June 2020, 13 September 2020, 5 June 2021, and 17 September 2021, respectively, for analysis of chemical and biological indicators. Soil total nitrogen was determined by the Kjeldahl method. Soil organic carbon was determined by the potassium dichromate capacity method and the external heating oxidation method.

Soil catalase activity was titrated by potassium permanganate, A total of 5.0 g of air-dried rhizosphere soil samples were added to 1 mL of toluene, then placed in a 4 °C freezer. After 30 min, a 3% $H_2O_2$ solution was added. The mixture was again placed at 4 °C and refrigerated for 30 min. Then, 0.5 mol L$^{-1}$ $H_2SO_4$ was quickly added, shaken, and filtered. The filtrate was added to 0.5 mol L$^{-1}$ $H_2SO_4$, and then titrated to light pink with 0.02 mol L$^{-1}$ KMnO$_4$. Leaving it on for 2 min without fading. Urease activity was performed by sodium phenolate-sodium hypochlorite colorimetry. An amount of 2 mL of toluene was added to 10 g of air-dried soil samples, and after 15 min, 10% urea solution and citrate buffer (pH = 6.7) were added to the mixture. The mixture was then incubated in a 37 °C incubator for 24 h. At the same time, soil-free control and matrix-free control were set. After 24 h of incubation, 38 °C distilled water was added and filtered, and then the filtrate was added to distilled water and sodium hypochlorite solution and measured at a wavelength of 578 nm using a spectrophotometer. Phosphatase activity was performed by disodium phosphate colorimetry. An amount of 5 g of air-dried soil samples were added to 1 mL of toluene, shaken well, and placed in a 37 °C incubator for 24 h. Then, 38 °C distilled water was added and filtered. The filtrate was added to borate buffer solution, potassium ferricyanide solution, and 4-aminoantipyrine solution to develop color and fill the volumetric flask. Finally, the reading was measured at 570 nm wavelength using a spectrophotometer.

Soil microorganism population density was determined by the dilution plate method. Soil bacteria, fungi, and actinomycetes were cultured in beef extract + peptone + agar medium. Martin medium and improved Gauss No. 1 medium, respectively. The specific operations were as follows: Weigh about 10 g of fresh rhizosphere soil sample, put it in 90 mL of sterile water, stir for 30 min at room temperature to obtain a soil solution with a concentration of $10^{-1}$, take 1 mL of the solution after standing, and add 9 mL of sterile water to obtain $10^{-2}$ diluent. By analogy, $10^{-3}$, $10^{-4}$, $10^{-5}$, and $10^{-6}$ diluents were obtained. Choose the right diluent according to the number of microorganisms in the soil. Fungi are between $10^{-1}$ and $10^{-3}$, actinomycetes between $10^{-3}$ and $10^{-5}$, and bacteria between $10^{-4}$ and $10^{-6}$.

### 2.3.2. Photosynthetic Characterization

Portable photosynthesis measurement systems (LI-6400, Li-Cor, Inc., Lincoln, NE, USA) were used to determine the photosynthetic properties from 9 am to 11:30 am under natural light conditions on 12 June 2020, 15 September 2020, 9 June 2021, and 13 September

2021. Three replicates were selected per treatment, and five fully extended young coffee leaves were selected for each replicate, fixed, and labeled. Then, clear and cloudless weather was selected to determine the photosynthetic characteristics of young coffee leaves. The daily mean values of net photosynthetic rate, transpiration rate, stomatal conductance, and intercellular $CO_2$ concentration were analyzed. Among them, the instantaneous water use efficiency of leaves was the ratio of net photosynthetic rate to transpiration rate.

### 2.4. Determination of Weights and Evaluation Indicators

#### 2.4.1. Determine Metric Weights

The entropy weight method was used to determine the weights of 15 indicators of soil organic carbon, organic matter, total nitrogen, carbon-to-nitrogen ratio, catalase, urease, phosphatase, bacteria, fungi, actinomycetes, net photosynthetic rate, transpiration rate, leaf water use efficiency, stomatal conductance, and intercellular $CO_2$ concentration [29], and the specific steps were as follows:

The first step was to calculate the standardized metric data. There were $n$ evaluation objects; $m$ evaluation indicators; the indicator data were standardized by Formula (2); $X_{ij}$ was the data of the $j$ index of the $i$ sample; and $A_{ij}$ indicates that the standardized data of the $j$ index of $i$ sample, using Equation (2), are:

$$A_{ij} = \frac{X_{ij} - \min\{X_{1j}, \cdots, X_{nj}\}}{\max\{X_{1j}, \cdots, X_{nj}\} - \min\{X_{1j}, \cdots, X_{nj}\}} \tag{2}$$

In the second step, the entropy value was calculated. Using Equation (3) to calculate the proportion of the ith sample value in the indicator under item $j$, $P_{ij}$ is:

$$P_{ij} = \frac{A_{ij}}{\sum\limits_{i=1}^{n} A_{ij}}, i = 1, 2, \ldots, n; \ j = 1, 2, \ldots, m \tag{3}$$

The entropy value $E_j$ for calculating the $j$th indicator is:

$$E_j = -\frac{1}{\ln(n)} \sum_{i=1}^{n} P_{ij} \ln(P_{ij}), j = 1, 2, \cdots, m \tag{4}$$

If $P_{ij} = 0$, then $\ln(P_{ij}) = 0$

The third step is to calculate the entropy weight. The weight $W_j$ of each indicator calculated by entropy is:

$$W_j = \frac{1 - E_j}{\sum\limits_{j=1}^{m} 1 - E_j}, (j = 1, 2, \cdots, m) \tag{5}$$

#### 2.4.2. TOPSIS Comprehensive Evaluation

In this study, 15 indexes of soil organic carbon, organic matter, total nitrogen, carbon-to-nitrogen ratio, catalase, urease, phosphatase, bacteria, fungi, actinomycetes, net photosynthetic rate, transpiration rate, leaf water use efficiency, stomatal conductance, and intercellular $CO_2$ concentration were selected to provide comprehensive scores for each treatment.

Normalizing the selected indexes,

$$Z_{ij} = \frac{X_{ij}}{\sqrt{\sum_{i=1}^{n} X^2_{ij}}} \tag{6}$$

where $Z_{ij}$ is the normalized value of $i$ treatment under $j$ index, $X_{ij}$ is the experimental value of $i$ treatment in $j$ index, $i = 1, 2 \ldots n$ ($n = 9$); $j = 1, 2 \ldots m$ ($m = 15$).

The positive ideal solution ($Z^+$) and the negative ideal result ($Z^-$) were obtained from the normalized results,

$$Z_{ij} = (z_{i1}{}^+, z_{i2}{}^+, z_{i3}{}^+ \cdots\cdots z_{i9}{}^+) \tag{7}$$

$$Z_{ij} = (z_{i1}{}^-, z_{i2}{}^-, z_{i3}{}^- \cdots\cdots z_{i9}{}^-) \tag{8}$$

where $Z_{ij}{}^+$ and $Z_{ij}{}^-$, respectively, represent the maximum value and minimum value of the evaluation object under the $j$ index in the normalized matrix.

The Euclidean distances of $D_i{}^+$ and $D_i{}^-$ between each index and positive and negative ideal solutions under each experimental treatment were calculated.

$$D_i^+ = \sqrt{\sum_{j=1}^{m}\left[w_j \times \left(z_{ij} - Z_{ij}^+\right)\right]^2} \tag{9}$$

$$D_i^- = \sqrt{\sum_{j=1}^{m}\left[w_j \times \left(z_{ij} - Z_{ij}^-\right)\right]^2} \tag{10}$$

where $u_j$ is the weight coefficient of evaluation index $j$, and its calculation method is shown in (3).

Calculating the relative approximation coefficient $R_i$ between different treatments and positive and negative ideal solutions.

$$R_i = D_i^- / (D_i^+ + D_i^-) \tag{11}$$

Each treatment (evaluation object) was sorted according to the $R_i$ value. The closer the $R_i$ value is to 1, it means that the treatment is an optimal treatment.

### 2.5. Statistical Analysis

Excel 2019 was used to collect and collate data, and Origin 2021 was used to analyze the Pearson correlation of the average value of the measured data, draw graphs, and perform principal component analysis on the means of 15 indicators across different treatments. Analysis of variance (ANOVA) was performed using IBM SPSS 21.0, and Duncan's multi-range test was used to compare whether there were significant differences between the treatments at the $p = 0.05$ level.

## 3. Results

### 3.1. Effects of Inorganic Fertilizer Level and Coffee Husk Returning Methods on Soil Fertility

Inorganic fertilizer level and coffee husk returning methods had significant effects on soil organic carbon, organic matter, total nitrogen, and carbon-to-nitrogen ratio ($p < 0.05$), but the interaction between the two had no significant effect on organic carbon and organic matter in June 2020, no significant effect on total nitrogen in September 2020 and September 2021, and the other effects were significant (Figure 1). From the average value of 2 years, significant differences were observed in soil organic carbon, organic matter, total nitrogen, and carbon-to-nitrogen ratio under different levels of inorganic fertilizer application for the same coffee husk returning methods. Specifically, the $F_M$ treatment showed higher soil organic carbon and organic matter content compared to the $F_H$ and FL treatments, with a trend of $F_M > F_H > F_L$. In comparison to the $F_L$ treatment, $F_M$ increased soil organic carbon, organic matter, and total nitrogen by 6.68%, 6.82%, and 6.12%, respectively. At the same level of inorganic fertilizer, significant differences were observed in soil organic carbon, organic matter, total nitrogen, and carbon-to-nitrogen ratio under different coffee husk returning methods. Compared with $C_A$, $C_C$ had the best effect on soil organic carbon, organic matter, and total nitrogen at the same level of inorganic fertilizer, increasing by 49.91%, 56.29%, and 32.65%, respectively, while $C_B$ had the greatest effect on the carbon-to-

nitrogen ratio by 20.42%. Among them, the content of soil organic carbon, organic matter, and total nitrogen was the highest under $F_MC_C$ treatment.

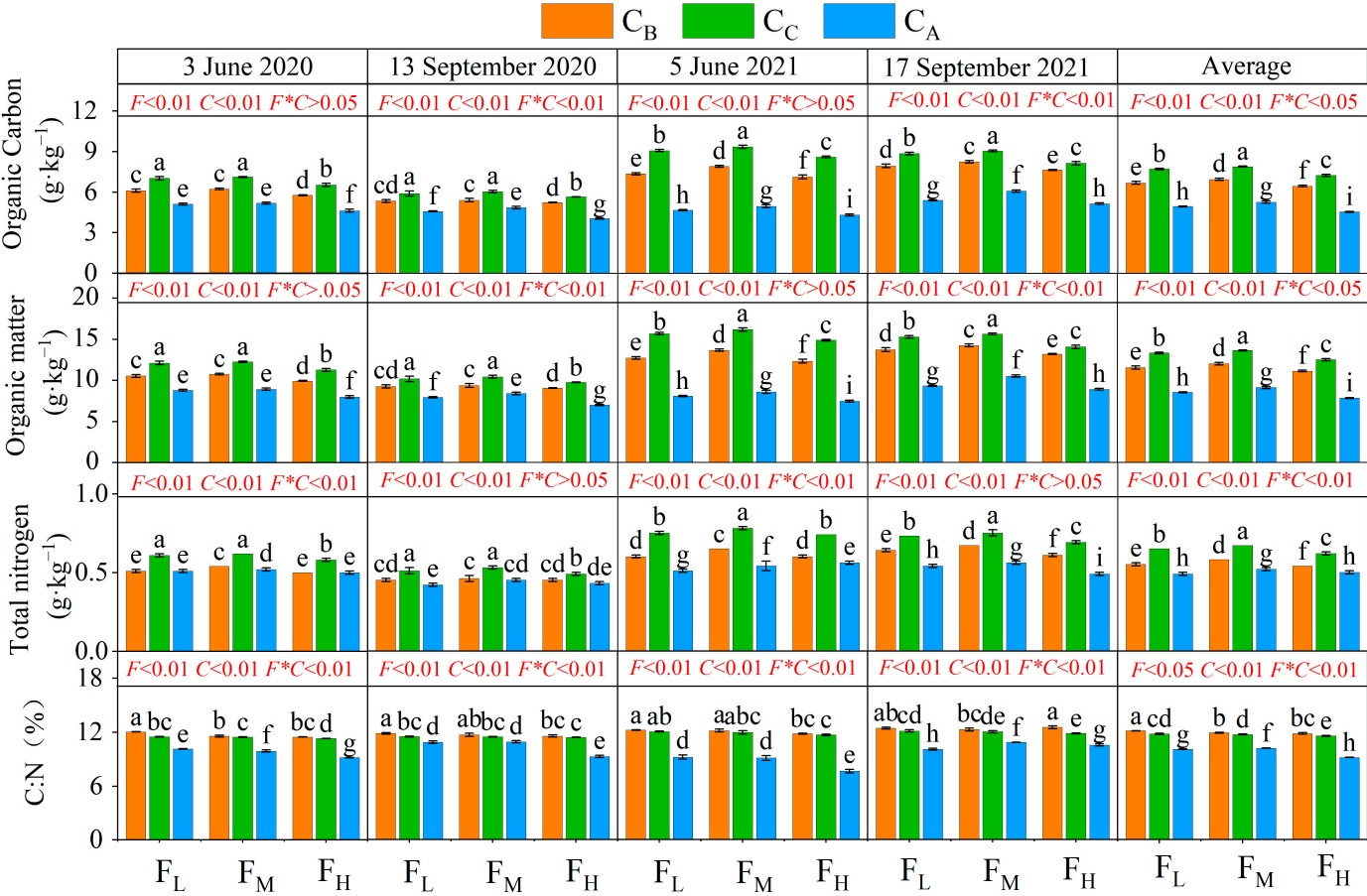

**Figure 1.** Effects of inorganic fertilizer level and coffee husk returning method on soil fertility. Note: $F_L$—low fertilizer; $F_M$—medium fertilizer; $F_H$—high fertilizer; $C_B$—coffee husk biochar; $C_C$—coffee husk compost; $C_A$—coffee husk ash; F—fertilization level; C—coffee husk returning method. Values are means ± standard errors (n = 3). Different letters in the same column indicate a significant difference at *p* < 0.05 level.

### 3.2. Effects of Inorganic Fertilizer Level and Coffee Husk Returning Methods on Soil Enzyme Activity

Inorganic fertilizer level and coffee husk returning methods had significant effects on soil catalase, phosphatase, and urease (*p* < 0.05), but the interaction between the two had no significant effect on the mean catalase values in September 2020, June 2021, and September 2021, and had no significant effect on urease in June 2021, and the rest had significant effects (Figure 2). From the average value of 2 years, significant differences were found in soil catalase, phosphatase, and urease activities under different levels of inorganic fertilizer application under the same coffee husk returning methods. Notably, the $F_M$ treatment showed significantly higher soil phosphatase and urease activities compared to the $F_L$ and $F_H$ treatments. Compared with $F_L$, $F_M$ increased soil phosphatase and urease activities by 13.82% and 20.00%, respectively, and $F_H$ decreased the activities of phosphatase and urease by 27.34% and 10.00%, respectively. Significant differences were found in soil catalase, phosphatase, and urease activities among different coffee husk returning methods at the same level of inorganic fertilizer application. Specifically, the soil phosphatase, urease, and catalase activities showed a trend of CC > CB > CA. Compared with $C_A$, the activities of $C_C$ on soil phosphatase and urease increased by 29.84% and 96.00%, respectively, and those of $C_B$ increased by 9.69% and 59.00%, respectively. In addition, the activities of soil

catalase, phosphatase, and urease were the largest under $F_M C_C$, which were 4.12, 420.89, and 76.60 mg $(g \cdot d)^{-1}$.

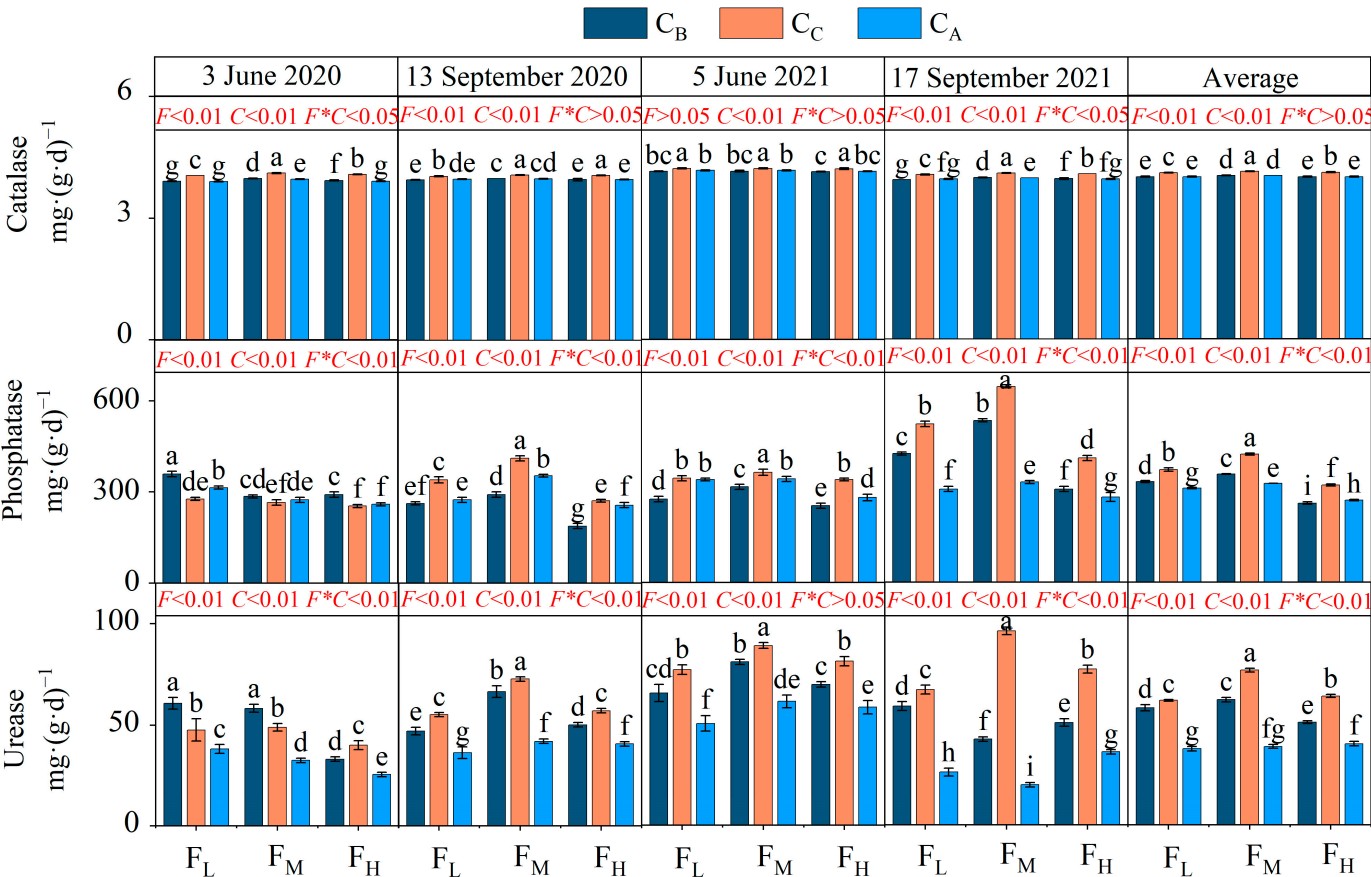

**Figure 2.** Effects of inorganic fertilizer level and coffee husk returning method on soil enzyme activity. Note: $F_L$—low fertilizer; $F_M$—medium fertilizer; $F_H$—high fertilizer; $C_B$—coffee husk biochar; $C_C$—coffee husk compost; $C_A$—coffee husk ash; F—fertilization level; C—coffee husk returning method. Values are means ± standard errors (n = 3). Different letters in the same column indicate a significant difference at the *p* < 0.05 level.

### 3.3. Effects of Inorganic Fertilizer Level and Coffee Husk Returning Methods on Soil Microorganisms

Inorganic fertilizer level, coffee husk returning methods, and the interaction between the two had significant effects on soil bacteria, fungi, and actinomycetes (Tables 1–3). Figure 3 shows the proportion of the three microorganisms. The mean value of the 2-year trial showed that the number of soil bacteria was significantly higher than that of fungi and actinomycetes, and the proportion of bacteria was the largest in all treatments. The effects of different coffee husk returning methods on soil microorganisms were different, among which the number of bacteria and actinomycetes under the $F_M$ treatment under $C_C$ was higher, while the number of fungi was greater under the $F_H$ treatment. The number of fungi and actinomycetes under $C_B$ treatment showed a trend of $F_M > F_L > F_H$, while bacteria were the largest $F_L$. After $C_A$ was returned to the field, bacteria, fungi, and actinomycetes showed the trend of $F_M > F_H > F_L$. In terms of the proportion of microbial numbers, the proportion of bacteria and actinomycetes under $F_M$ treatment was higher than that of other treatments, while the proportion of fungi was lower than that of other treatments. Compared with $C_A$, $C_C$ increased bacteria, fungi, and actinomycetes by 62.15%, 68.42%, and 46.21%, while $C_B$ increased by 40.50%, 10.86%, and 13.06%, respectively. It could be seen that $C_C$ t was more conducive to the growth and proliferation of soil bacteria, fungi, and actinomycetes under the same fertilization conditions. The proportion of bacteria,

fungi and actinomycetes varies in terms of the proportion of bacterial number, $C_B$ and $C_C$ were higher than $C_A$, while the proportion of fungal number was the largest in $C_C$.

**Table 1.** Effects of inorganic fertilizer level and coffee husk returning method on soil bacterial population ($\times 10^5/CFU \cdot g^{-1}$).

| Fertilization Level | Coffee Husk Returning Method | Sampling Time | | | | Average |
|---|---|---|---|---|---|---|
| | | **3 June 2020** | **13 September 2020** | **5 June 2021** | **17 September 2021** | |
| $F_L$ | $C_B$ | $9.51 \pm 2.35$ de | $11.97 \pm 1.04$ a | $12.23 \pm 1.68$ bc | $15.97 \pm 2.2$ bc | $12.42 \pm 1.82$ bcd |
| | $C_C$ | $13.89 \pm 0.41$ abc | $9.55 \pm 0.10$ b | $13.20 \pm 0.25$ b | $17.23 \pm 0.33$ b | $13.47 \pm 0.208$ b |
| | $C_A$ | $8.94 \pm 3.67$ def | $5.86 \pm 0.31$ e | $8.93 \pm 0.16$ efg | $11.61 \pm 0.88$ efg | $8.84 \pm 1.16$ fg |
| $F_M$ | $C_B$ | $10.21 \pm 1.23$ de | $9.18 \pm 1.65$ bc | $11.18 \pm 0.30$ cd | $14.61 \pm 0.19$ cd | $11.30 \pm 0.54$ cde |
| | $C_C$ | $16.73 \pm 1.28$ ab | $12.58 \pm 4.11$ a | $16.11 \pm 1.86$ a | $21.05 \pm 2.45$ a | $16.62 \pm 2.001$ a |
| | $C_A$ | $10.68 \pm 2.56$ cde | $6.77 \pm 0.01$ de | $10.23 \pm 0.43$ de | $13.34 \pm 0.65$ de | $10.25 \pm 0.69$ ef |
| $F_H$ | $C_B$ | $8.29 \pm 0.49$ efg | $4.97 \pm 0.15$ e | $8.16 \pm 0.30$ gh | $10.61 \pm 0.40$ g | $8.01 \pm 0.33$ gh |
| | $C_C$ | $12.23 \pm 2.41$ bcd | $6.71 \pm 0.26$ de | $10.97 \pm 0.29$ cd | $14.30 \pm 0.32$ cd | $11.06 \pm 0.65$ de |
| | $C_A$ | $7.32 \pm 1.73$ efg | $6.74 \pm 0.31$ de | $8.56 \pm 0.88$ fg | $11.14 \pm 1.15$ fg | $8.44 \pm 0.95$ fg |
| Significance test | | | | | | |
| | F | ** | ** | ** | ** | ** |
| | C | ** | ** | ** | ** | ** |
| | F*C | ** | ** | ** | ** | ** |

Note: $F_L$—low fertilizer; $F_M$—medium fertilizer; $F_H$—high fertilizer; $C_B$—coffee husk biochar; $C_C$—coffee husk compost; $C_A$—coffee husk ash; F—fertilization level; C—coffee husk returning method. Values are means ± standard errors (n = 3). Different letters in the same column indicate significant difference at $p < 0.05$ level. ** means an extremely significant difference ($p < 0.01$).

**Table 2.** Effects of inorganic fertilizer level and coffee husk returning method on soil fungus population ($\times 10^5/CFU\ g^{-1}$).

| Fertilization Level | Coffee Husk Returning Method | Sampling Time | | | | Average |
|---|---|---|---|---|---|---|
| | | **3 June 2020** | **13 September 2020** | **5 June 2021** | **17 September 2021** | |
| $F_L$ | $C_B$ | $4.25 \pm 0.34$ cde | $2.48 \pm 0.01$ def | $4.01 \pm 0.34$ cd | $6.83 \pm 0.92$ bc | $4.39 \pm 0.40$ cd |
| | $C_C$ | $4.57 \pm 0.09$ c | $3.23 \pm 0.03$ c | $4.27 \pm 0.09$ c | $6.22 \pm 0.23$ cd | $4.57 \pm 0.10$ c |
| | $C_A$ | $3.85 \pm 0.62$ de | $2.34 \pm 0.05$ def | $3.63 \pm 0.61$ de | $6.01 \pm 1.63$ cde | $3.96 \pm 0.72$ def |
| $F_M$ | $C_B$ | $4.28 \pm 0.04$ cd | $2.84 \pm 0.04$ cd | $4.01 \pm 0.04$ cd | $6.20 \pm 0.18$ cd | $4.33 \pm 0.06$ cd |
| | $C_C$ | $5.86 \pm 1.01$ b | $4.58 \pm 1.26$ b | $5.43 \pm 0.89$ b | $7.11 \pm 0.28$ b | $5.74 \pm 0.86$ b |
| | $C_A$ | $4.14 \pm 0.13$ cde | $3.09 \pm 0.11$ c | $3.86 \pm 0.13$ cde | $5.33 \pm 0.14$ def | $4.10 \pm 0.13$ cde |
| $F_H$ | $C_B$ | $3.67 \pm 0.02$ ef | $2.69 \pm 0.02$ cde | $3.42 \pm 0.01$ ef | $4.81 \pm 0.01$ fg | $3.65 \pm 0.01$ efg |
| | $C_C$ | $7.15 \pm 0.25$ a | $5.48 \pm 0.20$ a | $6.64 \pm 0.23$ a | $8.91 \pm 0.28$ a | $7.04 \pm 0.24$ a |
| | $C_A$ | $4.12 \pm 0.02$ cde | $2.71 \pm 0.03$ cde | $3.86 \pm 0.02$ cde | $6.02 \pm 0.05$ cde | $4.18 \pm 0.02$ cde |
| Significance test | | | | | | |
| | F | ** | ** | ** | ** | ** |
| | C | ** | ** | ** | ** | ** |
| | F*C | ** | ** | ** | ** | ** |

Note: $F_L$—low fertilizer; $F_M$—medium fertilizer; $F_H$—high fertilizer; $C_B$—coffee husk biochar; $C_C$—coffee husk compost; $C_A$—coffee husk ash; F—fertilization level; C—coffee husk returning method. Values are means ± standard errors (n = 3). Different letters in the same column indicate significant difference at $p < 0.05$ level. ** means an extremely significant difference ($p < 0.01$).

**Table 3.** Effects of inorganic fertilizer level and coffee husk returning method on soil actinomycetes population ($\times 10^5$/CFU·g$^{-1}$).

| Fertilization Level | Coffee Husk Returning Method | Sampling Time | | | | Average |
|---|---|---|---|---|---|---|
| | | 3 June 2020 | 13 September 2020 | 5 June 2021 | 17 September 2021 | |
| F$_L$ | C$_B$ | 3.01 ± 0.47 bcd | 2.11 ± 0.44 fg | 4.17 ± 0.67 def | 4.17 ± 0.74 def | 3.36 ± 0.58 defg |
| | C$_C$ | 4.37 ± 0.23 ab | 1.79 ± 1.42 gh | 4.70 ± 1.13 de | 4.73 ± 1.27 de | 3.90 ± 0.90 de |
| | C$_A$ | 2.09 ± 0.25 de | 3.08 ± 0.82 ef | 4.47 ± 0.83 de | 4.53 ± 0.92 de | 3.54 ± 0.69 def |
| F$_M$ | C$_B$ | 3.39 ± 0.20 bcd | 4.25 ± 0.51 c | 6.33 ± 0.49 bc | 6.55 ± 0.54 bc | 5.13 ± 0.40 bc |
| | C$_C$ | 5.68 ± 0.29 a | 7.09 ± 0.27 a | 10.27 ± 0.30 a | 10.88 ± 0.33 a | 8.48 ± 0.26 a |
| | C$_A$ | 2.93 ± 0.80 bcd | 5.53 ± 0.63 b | 7.20 ± 0.97 b | 7.54 ± 1.06 b | 5.80 ± 0.84 b |
| F$_H$ | C$_B$ | 2.21 ± 0.92 cde | 1.46 ± 0.19 gh | 3.10 ± 0.70 fgh | 2.99 ± 0.76 fgh | 2.44 ± 0.63 fgh |
| | C$_C$ | 3.82 ± 0.97 bc | 2.90 ± 0.64 ef | 5.37 ± 0.02 cd | 5.48 ± 0.01 cd | 4.39 ± 0.09 cd |
| | C$_A$ | 2.55 ± 2.88 cde | 2.83 ± 0.39 ef | 4.53 ± 1.71 de | 4.59 ± 1.81 de | 3.62 ± 1.58 de |
| Significance test | | | | | | |
| F | | ** | ** | ** | ** | ** |
| C | | ** | ** | ** | ** | ** |
| F*C | | ns | ** | ** | ** | ** |

Note: F$_L$—low fertilizer; F$_M$—medium fertilizer; F$_H$—high fertilizer; C$_B$—coffee husk biochar; C$_C$—coffee husk compost; C$_A$—coffee husk ash; F—fertilization level; C—coffee husk returning method. Values are means ± standard errors (n = 3). Different letters in the same column indicate significant difference at $p < 0.05$ level. ** means an extremely significant difference ($p < 0.01$), and ns means no significant difference ($p > 0.05$).

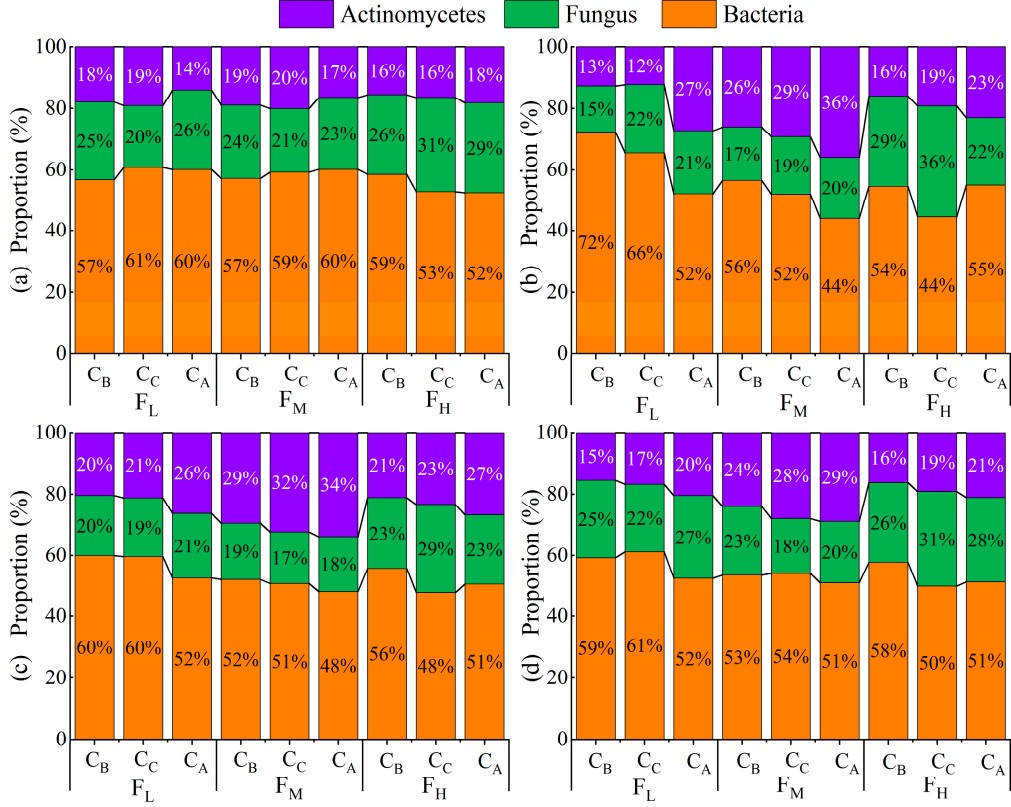

**Figure 3.** The proportion of microorganisms under different inorganic fertilizer levels and the coffee husk returning method. Note: (**a**) 3 June 2020; (**b**) 13 September 2020; (**c**) 5 June 2021; (**d**) 17 September 2021. F$_L$—low fertilizer; F$_M$—medium fertilizer; F$_H$—high fertilizer; C$_B$—coffee husk biochar; C$_C$—coffee husk compost; C$_A$—coffee husk ash. Values are means ± standard errors (n = 3).

### 3.4. Effects of Inorganic Fertilizer Level and Coffee Husk Returning Methods on Photosynthetic Characteristics of Arabica Coffee

Table 4 showed that the inorganic fertilizer level and coffee husk returning methods had significant effects on the mean values of Pn, Tr, Gs, Ci, and LWUE at 2 years ($p < 0.05$), but the interaction between the two only had a significant effect on Tr, Gs, and Ci. From the average value of 2 years, the effects of different fertilization rates on leaves Pn, Tr, Gs, and LWUE showed the trend of $F_M > F_H > F_L$, while Ci showed the trend of $F_H > F_M > F_L$. Under the same inorganic fertilizer level, compared with $C_B$, $C_C$, and $C_A$ had different degrees of improvement for Pn, Tr, and Gs, while LWUE and Ci decreased slightly, among which $C_C$ increased Pn, Tr, and Gs by 19.33%, 27.99%, and 83.33%, respectively, and decreased by 11.85% and 8.56% for LWUE and Ci. In addition, Pn, Tr, and Gs are the largest under $F_M C_C$ processing, and Ci is also the lowest.

### 3.5. Correlation Analysis of Soil Ecological Environment and Coffee Photosynthetic Characteristics

In 2020, there was a positive correlation between soil TOC, SOM, TN, C/N, CA, PS, UA, BE, FI, AN, Pn, and LWUE, and the correlation was significant ($p < 0.05$) and negatively correlated with Tr, Gs, and Ci. Among them, the correlation coefficient between TOC and SOM was the largest, and the correlation coefficient between Pn and BE was the smallest, with correlation coefficients of 1 and 0.042, respectively. The overall change in 2021 was similar to that in 2020, but LWUE was negatively correlated with soil TOC, SOM, TN, C/N, CA, PS, UA, BE, FI, and AN, with a minimum correlation coefficient of $-0.66$ (Figure 4).

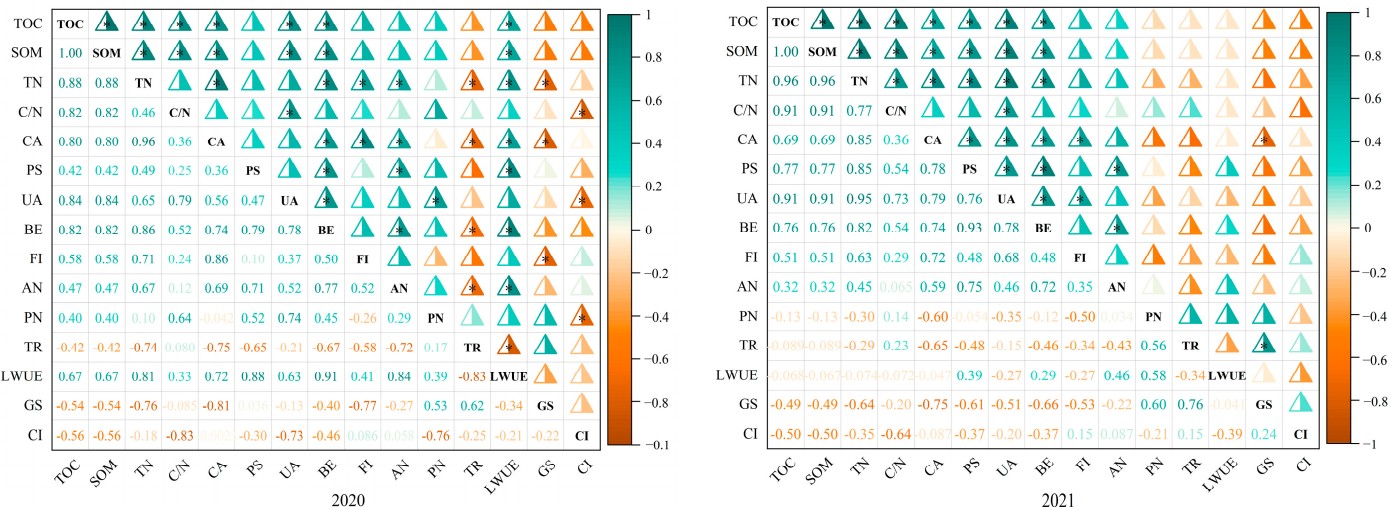

**Figure 4.** Correlation analysis of the soil ecological environment and coffee photosynthetic characteristics. Note: * means a significant difference ($p < 0.05$), respectively. TOC—organic carbon; SOM—organic matter; TN—total nitrogen; C/N—carbon-to-nitrogen ratio; CA—catalase; PS—phosphatase; UA—urease; BE—bacteria; FI—fungi; AN—actinomycetes; Pn—net photosynthetic rate; Gs—stomatal conductance; Tr—transpiration rate; LWUE—leaf water use efficiency; Ci—intercellular $CO_2$ concentration.

**Table 4.** Effects of inorganic fertilizer level and coffee husk returning method on photosynthetic indexes of Arabica coffee.

| Sampling Time | Photosynthetic Indicators | Sampling Time | | | | | | | | | Significance Test | | |
|---|---|---|---|---|---|---|---|---|---|---|---|---|---|
| | | $F_L C_B$ | $F_L C_C$ | $F_L C_A$ | $F_M C_B$ | $F_M C_C$ | $F_M C_A$ | $F_H C_B$ | $F_H C_C$ | $F_H C_A$ | F | C | F*C |
| 12 June 2020 | Pn ($\mu$mol m$^{-2}$ s$^{-1}$) | 9.36 ± 0.47 bcd | 10.15 ± 0.60 ab | 9.11 ± 0.23 cd | 9.79 ± 0.29 abc | 10.55 ± 0.43 a | 9.67 ± 0.64 abc | 8.55 ± 0.65 d | 9.22 ± 0.43 cd | 8.57 ± 0.45 d | ** | ** | ns |
| | Tr (mmol m$^{-2}$ s$^{-1}$) | 3.68 ± 0.06 d | 4.25 ± 0.01 a | 3.79 ± 0.05 cd | 3.25 ± 0.03 e | 4.25 ± 0.06 a | 3.86 ± 0.16 c | 3.85 ± 0.14 c | 4.34 ± 0.01 a | 4.08 ± 0.05 b | ** | ** | ** |
| | LWUE ($\mu$mol mmol$^{-1}$) | 2.54 ± 0.13 cd | 2.39 ± 0.14 bc | 2.40 ± 0.03 bc | 3.01 ± 0.09 a | 2.48 ± 0.12 b | 2.51 ± 0.25 b | 2.22 ± 0.09 b | 2.12 ± 0.10 d | 2.10 ± 0.13 d | ** | ** | ns |
| | Gs (mol m$^{-2}$ s$^{-1}$) | 0.08 ± 0.01 b | 0.12 ± 0.01 a | 0.12 ± 0.01 a | 0.09 ± 0.02 b | 0.13 ± 0.01 a | 0.13 ± 0.01 a | 0.06 ± 0.01 c | 0.09 ± 0.01 b | 0.10 ± 0.01 b | ** | ** | ns |
| | Ci ($\mu$mol·mol$^{-1}$) | 95.04 ± 9.15 cd | 82.39 ± 10.27 d | 116.14 ± 3.5 bc | 126.2 ± 12.01 ab | 116.11 ± 7.48 bc | 136.65 ± 25.35 ab | 125.5 ± 16.84 ab | 117.78 ± 12.59 bc | 144.48 ± 4.85 a | ** | ** | ns |
| 15 September 2020 | Pn ($\mu$mol m$^{-2}$ s$^{-1}$) | 6.9 ± 0.55 abc | 7.84 ± 0.49 ab | 6.79 ± 1.04 abc | 7.34 ± 0.54 abc | 8.12 ± 0.57 a | 6.73 ± 0.81 bc | 6.24 ± 0.64 c | 6.86 ± 0.46 bc | 6.01 ± 1.01 c | * | ** | ns |
| | Tr (mmol m$^{-2}$ s$^{-1}$) | 1.97 ± 0.10 d | 2.88 ± 0.24 bd | 2.43 ± 0.41 bcd | 1.92 ± 0.05 d | 2.74 ± 0.13 bc | 2.35 ± 0.37 bcd | 2.19 ± 0.27 cd | 3.46 ± 0.35 a | 2.57 ± 0.54 bc | * | * | ns |
| | LWUE ($\mu$mol mmol$^{-1}$) | 3.50 ± 0.20 b | 2.73 ± 0.12 c | 2.79 ± 0.05 c | 3.82 ± 0.26 a | 2.96 ± 0.21 c | 2.88 ± 0.23 c | 2.85 ± 0.07 c | 1.99 ± 0.14 e | 2.35 ± 0.10 d | ** | ** | ns |
| | Gs (mol m$^{-2}$ s$^{-1}$) | 0.05 ± 0.01 d | 0.16 ± 0.02 a | 0.13 ± 0.01 bc | 0.06 ± 0.01 d | 0.15 ± 0.01 ab | 0.13 ± 0.02 bc | 0.05 ± 0.02 d | 0.13 ± 0.03 bc | 0.11 ± 0.02 c | * | ** | ns |
| | Ci ($\mu$mol·mol$^{-1}$) | 238.82 ± 7.22 b | 173.66 ± 13.36 d | 266.88 ± 11.39 a | 220.76 ± 11.65 bc | 200.87 ± 30.82 c | 267.41 ± 11.22 a | 247.16 ± 12.5 ab | 234.33 ± 4.65 b | 269.21 ± 11.97 a | ** | ** | * |
| 9 June 2021 | Pn ($\mu$mol m$^{-2}$ s$^{-1}$) | 4.98 ± 0.31 de | 7.51 ± 0.64 a | 5.84 ± 0.37 bcd | 5.16 ± 0.75 de | 8.33 ± 0.47 a | 6.51 ± 0.50 b | 4.82 ± 0.26 e | 6.32 ± 0.53 bc | 5.54 ± 0.39 cde | ** | ** | ns |
| | Tr (mmol m$^{-2}$ s$^{-1}$) | 2.52 ± 0.09 d | 3.71 ± 0.15 a | 2.67 ± 0.18 cd | 2.67 ± 0.33 cd | 4.01 ± 0.26 a | 3.02 ± 0.22 bc | 2.88 ± 0.15 bcd | 3.74 ± 0.23 a | 3.14 ± 0.09 b | ** | ** | ns |
| | LWUE ($\mu$mol mmol$^{-1}$) | 1.98 ± 0.17 bc | 2.02 ± 0.21 abc | 2.19 ± 0.06 a | 1.92 ± 0.08 cd | 2.08 ± 0.13 abc | 2.15 ± 0.02 ab | 1.68 ± 0.04 e | 1.69 ± 0.04 e | 1.76 ± 0.08 de | ** | * | ns |
| | Gs (mol m$^{-2}$ s$^{-1}$) | 0.03 ± 0.01 f | 0.09 ± 0.02 cd | 0.08 ± 0.02 cd | 0.07 ± 0.01 c | 0.12 ± 0.01 b | 0.11 ± 0.02 bc | 0.09 ± 0.01 cd | 0.17 ± 0.02 a | 0.13 ± 0.02 b | ** | ** | ns |
| | Ci ($\mu$mol·mol$^{-1}$) | 151.42 ± 9.66 d | 127.17 ± 5.54 d | 173.14 ± 3.27 c | 185.93 ± 10.46 c | 180.83 ± 7.84 c | 190.68 ± 3.43 c | 212.19 ± 16.34 b | 180.51 ± 14.97 c | 245.86 ± 7.52 a | ** | ** | ** |
| 13 September 2021 | Pn ($\mu$mol m$^{-2}$ s$^{-1}$) | 4.67 ± 0.2d c | 5.03 ± 0.29 c | 4.85 ± 0.63 c | 5.23 ± 0.26 bc | 5.85 ± 0.10 a | 5.73 ± 0.42 ab | 4.23 ± 0.21 d | 5.04 ± 0.31 c | 5.03 ± 0.09 c | ** | ** | ns |
| | Tr (mmol m$^{-2}$ s$^{-1}$) | 2.05 ± 0.10 b | 2.26 ± 0.24 b | 2.08 ± 0.22 b | 2.19 ± 0.20 b | 2.13 ± 0.09 b | 2.06 ± 0.14 b | 2.32 ± 0.09 b | 2.84 ± 0.25 a | 2.72 ± 0.10 a | ** | ** | ns |
| | LWUE ($\mu$mol mmol$^{-1}$) | 2.29 ± 0.12 b | 2.24 ± 0.13 b | 2.34 ± 0.07 b | 2.40 ± 0.11 b | 2.75 ± 0.07 a | 2.79 ± 0.11 a | 1.83 ± 0.02 c | 1.78 ± 0.06 c | 1.85 ± 0.04 c | ** | ** | ** |
| | Gs (mol m$^{-2}$ s$^{-1}$) | 0.02 ± 0.01 c | 0.06 ± 0.02 ab | 0.08 ± 0.02 a | 0.06 ± 0.02 ab | 0.07 ± 0.02 a | 0.07 ± 0.01 a | 0.03 ± 0.01 c | 0.06 ± 0.01 ab | 0.04 ± 0.01 bc | ** | ** | ns |
| | Ci ($\mu$mol·mol$^{-1}$) | 215.42 ± 6.1 d | 243.22 ± 5.83 b | 216.43 ± 6.36 d | 223.8 ± 12.16 cd | 221.08 ± 5.86 cd | 240.6 ± 5.52 b | 232.61 ± 7.81 b | 217.14 ± 5.55 d | 262.52 ± 8.65 a | ** | ** | ** |
| Average | Pn ($\mu$mol m$^{-2}$ s$^{-1}$) | 6.48 ± 0.06 de | 7.63 ± 0.43 b | 6.65 ± 0.37 de | 6.88 ± 0.42 cd | 8.21 ± 0.12 a | 7.16 ± 0.19 c | 5.96 ± 0.10 f | 6.86 ± 0.16 cd | 6.29 ± 0.06 ef | ** | ** | ns |
| | Tr (mmol m$^{-2}$ s$^{-1}$) | 2.56 ± 0.02 d | 3.27 ± 0.06 b | 2.74 ± 0.16 c | 2.51 ± 0.13 d | 3.28 ± 0.06 b | 2.83 ± 0.03 c | 2.81 ± 0.09 c | 3.60 ± 0.14 a | 3.13 ± 0.12 b | ** | ** | * |
| | LWUE ($\mu$mol mmol$^{-1}$) | 2.58 ± 0.03 b | 2.35 ± 0.07 c | 2.43 ± 0.03 c | 2.79 ± 0.07 a | 2.57 ± 0.07 b | 2.59 ± 0.02 b | 2.14 ± 0.04 e | 1.90 ± 0.05 f | 2.02 ± 0.07 d | ** | ** | ns |
| | Gs (mol m$^{-2}$ s$^{-1}$) | 0.05 ± 0.01 e | 0.11 ± 0.01 abc | 0.10 ± 0.01 bc | 0.07 ± 0.00 d | 0.12 ± 0.01 a | 0.11 ± 0.01 a | 0.06 ± 0.01 d | 0.11 ± 0.00 ab | 0.10 ± 0.01 c | ** | ** | * |
| | Ci ($\mu$mol·mol$^{-1}$) | 175.17 ± 1.64 e | 156.61 ± 1.74 f | 193.15 ± 5.20 c | 189.17 ± 7.89 cd | 179.72 ± 5.77 de | 208.84 ± 8.23 b | 204.36 ± 6.47 b | 187.44 ± 4.52 cd | 230.52 ± 3.04 a | ** | ** | * |

Note: $F_L$—low fertilizer; $F_M$—medium fertilizer; $F_H$—high fertilizer; $C_B$—coffee husk biochar; $C_C$—coffee husk compost; $C_A$—coffee husk ash; Pn—net photosynthetic rate; Gs—stomatal conductance; Tr—transpiration rate; LWUE—leaf water use efficiency; Ci—intercellular $CO_2$ concentration; F—fertilization level; C—coffee husk returning method. Values are means ± standard errors (n = 3). Different letters in the same column indicate a significant difference at $p < 0.05$ level. * means a significant difference ($p < 0.05$), ** means an extremely significant difference ($p < 0.01$), and ns means no significant difference ($p > 0.05$).

### 3.6. Principal Component Analysis of Different Inorganic Fertilizer Levels and Coffee Husk Returning Methods

Principal component analysis is a statistical method that utilizes dimensionality reduction to transform multiple indicators into a few comprehensive indicators that retain most of the information of the original indicators and are not related to each other. This study selected the mean values of 15 indicators, including soil organic carbon, organic matter, total nitrogen, carbon-to-nitrogen ratio, catalase, urease, phosphatase, bacteria, fungi, actinomycetes, net photosynthetic rate, transpiration rate, leaf water use efficiency, stomatal conductance, and intercellular $CO_2$ concentration, for principal component analysis. The results are shown in Figure 5. The results of the principal component analysis showed that the first principal component had a contribution rate of 55.84%, with main factors including catalase, urease, phosphatase, bacteria, fungi, actinomycetes, organic matter, total nitrogen, and carbon-to-nitrogen ratio. The second principal component had a contribution rate of 19.58%, with main factors such as net photosynthetic rate, transpiration rate, leaf water use efficiency, carbon-to-nitrogen ratio, and stomatal conductance.

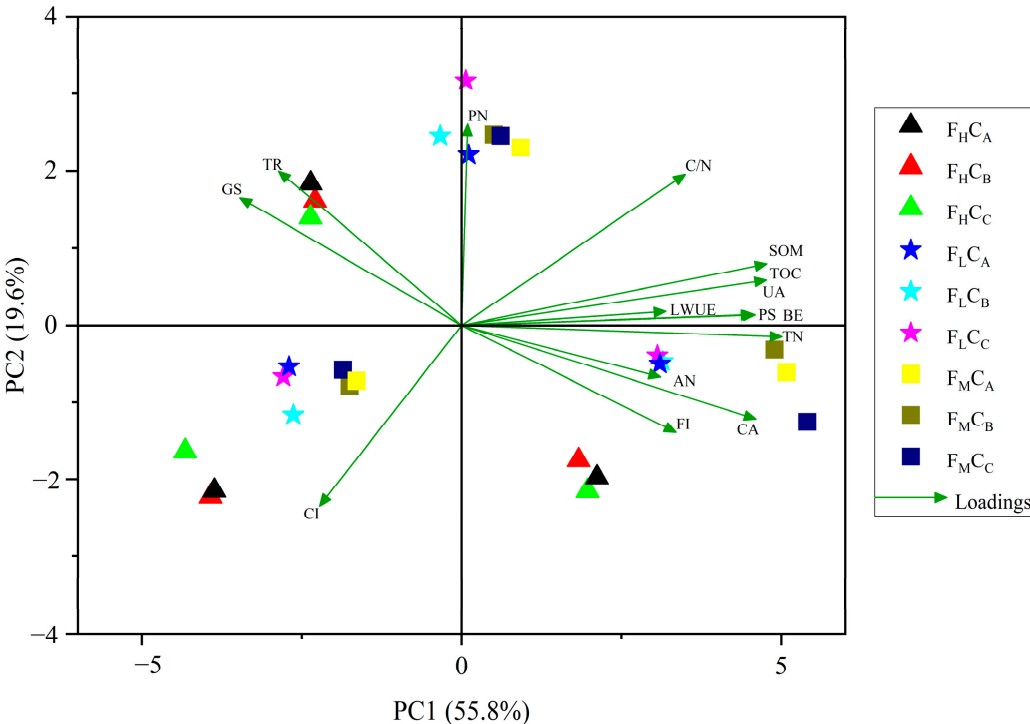

**Figure 5.** Principal component analysis (PCA) of various indicators under different treatments. Note: $F_L$—low fertilizer; $F_M$—medium fertilizer; $F_H$—high fertilizer; $C_B$—coffee husk biochar; $C_C$—coffee husk compost; $C_A$—coffee husk ash; TOC—organic carbon; SOM—organic matter; TN—total nitrogen; C/N—carbon-to-nitrogen ratio; CA—catalase; PS—phosphatase; UA—urease; BE—bacteria; FI—fungi; AN—actinomycetes; Pn—net photosynthetic rate; Gs—stomatal conductance; Tr—transpiration rate; LWUE—leaf water use efficiency; Ci—intercellular $CO_2$ concentration.

### 3.7. Comprehensive Evaluation of Inorganic Fertilizer Level and Coffee Husk Returning Methods Based on the Entropy Weight Method—TOPSIS Method

The entropy weight method was used to determine the weights of the mean soil fertility and photosynthetic indexes over 2 years. Among them, the weights of CA, PS, UA, BE, FI, AN, TOC, SOM, TN, C/N, Pn, Tr, Gs, Ci, and LWUE are 0.13, 0.06, 0.08, 0.08, 0.09, 0.07, 0.05, 0.05, 0.07, 0.04, 0.06, 0.07, 0.05, 0.04, and 0.05, respectively. According to the TOPSIS method, the comprehensive benefit evaluation of each treatment showed that the top 3 scoring results were $F_M C_C$, $F_H C_C$, and $F_M C_B$, among which the highest proximity was 0.75 and the lowest was 0.23. The results showed that the relative proximity coefficient

of $F_MC_C$ was close to 1, and the comprehensive score method showed that $F_MC_C$ was the optimal treatment combination (Table 5).

**Table 5.** Comprehensive evaluation and ranking results of inorganic fertilizer levels and the coffee husk returning method by the TOPSIS method.

| Fertilization Level | Coffee Husk Returning Method | Euclidean Distance | | Relative Coefficient | Rank |
|---|---|---|---|---|---|
| | | $D_i^+$ | $D_i^-$ | | |
| $F_L$ | $C_B$ | 0.03 | 0.02 | 0.40 | 5 |
| $F_L$ | $C_C$ | 0.03 | 0.03 | 0.43 | 4 |
| $F_L$ | $C_A$ | 0.04 | 0.01 | 0.22 | 9 |
| $F_M$ | $C_B$ | 0.03 | 0.03 | 0.49 | 3 |
| $F_M$ | $C_C$ | 0.02 | 0.05 | 0.75 | 1 |
| $F_M$ | $C_A$ | 0.04 | 0.02 | 0.38 | 6 |
| $F_H$ | $C_B$ | 0.05 | 0.02 | 0.27 | 7 |
| $F_H$ | $C_C$ | 0.03 | 0.03 | 0.51 | 2 |
| $F_H$ | $C_A$ | 0.04 | 0.01 | 0.23 | 9 |

Note: $F_L$—low fertilizer; $F_M$—medium fertilizer; $F_H$—high fertilizer; $C_B$—coffee husk biochar; $C_C$—coffee husk compost; $C_A$—coffee husk ash.

## 4. Discussion

### 4.1. Effects of Inorganic Fertilizer Level and Coffee Husk Returning Methods on Soil Fertility

Appropriate drip irrigation and fertilization practices are crucial to maintaining loose soil structure and regulating the levels of total nitrogen, organic carbon, and organic matter [9,30]. The findings demonstrated that medium fertilizer ($F_M$) significantly increased soil organic carbon and organic matter, consistent with previous research [31]. Conversely, high fertilizer ($F_H$) led to declining soil organic matter. Presumably, the excessive application of chemical fertilizer caused nitrogen to be converted into nitrate, which complexed with soil organic matter to form other chemical substances that decreased the content of organic matter [13].

In this study, it was revealed that the effects of three different coffee husk returning methods on soil fertility were different. The results demonstrated that $C_C$ significantly increased the soil organic matter, organic carbon, and total nitrogen content compared to $C_B$ and $C_A$. This might be due to the high humus content of $C_C$, which contained a large amount of organic carbon and organic nitrogen that directly improved soil fertility upon application. Additionally, the compost exhibited a significant amount of macromolecular complexes. Furthermore, it was abundant in multivalent ions such as calcium and phosphorus, which could act as a bridge and enhance the adsorption potential of organic matter on minerals [32,33]. They aided the attachment, growth, and reproduction of microorganisms, fostered the establishment of rich microbial communities, promoted the formation of organic carbon, and expanded the effective carbon pool of microorganisms [6]. While $C_B$ and $C_A$ had high losses of organic carbon and nitrogen, up to 90%, during the treatment processing, which weakened their effective nutrient content [10]. Furthermore, another study revealed that biochar application reduced the concentration of ammonium and nitrate in the soil [16], resulting in a decrease in total nitrogen. The carbon-to-nitrogen ratio is a crucial factor affecting the rate of soil mineralization. $C_B$ had the highest increase in the carbon-to-nitrogen ratio, mainly because biochar itself was rich in carbon and the treatment had a large loss of nitrogen.

### 4.2. Effects of Inorganic Fertilizer Level and Coffee Husk Returning Methods on Soil Microbial Activity

Soil microorganisms play a vital role in soil structure formation as well as organic matter and mineral decomposition [34]. Previous research has shown that appropriate fertilization levels promote the proliferation of soil bacteria, fungi, and actinomycetes,

mainly due to the nitrogen, phosphorus, and potassium provided by inorganic fertilizers, which serve as important nutrient sources for the growth and reproduction of soil microorganisms [5,35]. However, excessive fertilization led to a reduction in bacterial activity and diversity [35], consistent with our findings. Other studies have indicated that with increasing nitrogen application, the fungal biomass decreased while the bacterial biomass remained relatively stable [36,37], which differed from our research results. This might be because excessive fertilization inhibited the growth and reproduction of bacteria and actinomycetes and allowed fungi to gain a competitive advantage, leading to an increase in their amount. In addition, fungi may have a stronger advert resistance. The increase in the amount of fungi in $C_C$ was primarily because the compost products served as an excellent environment for fungal colonization, and the bonding of fungal hyphae to form large aggregates promoted soil aggregation, which further provided diversified habitats for fungi, resulting in increased fungal activity and diversity [6]. While the increase in bacterial amounts might be because the compost provided a large amount of unstable carbon sources for microorganisms to produce extracellular polymeric substances, which promoted bacterial proliferation [31]. The combined application of compost and chemical fertilizers has been shown to significantly increase the amount of bacteria, fungi, and actinomycetes in the rhizosphere [38], which is consistent with our findings.

### 4.3. Effects of Inorganic Fertilizer Level and Coffee Husk Returning Methods on Soil Enzyme Activity

Soil enzymes come from microbial activities, plant root exudates, and the decomposition of animal and plant residues. Their activities are closely related to nutrients and microbial activities, and thus soil enzymes are important indicators for evaluating soil quality and sensitivity. This study observed a pattern of first an increase and then a decrease in the activities of the three enzymes with increasing fertilization application. This was because high fertilizer input resulted in increased soil osmotic pressure, inhibited root growth, and reduced the amount of root exudates, which led to dehydration and the death of microorganisms and ultimately decreased enzyme activity. The activities of urease and phosphatase were higher for $C_C$. On the one hand, it could be attributed to the fact that manure is rich in organic matter, which produces a large number of enzymes during the decomposition process of microorganisms, and these microorganisms also release active enzymes after apoptosis [39]. On the other hand, the decomposition of compost produced phytohormones that promoted root growth. At the same time, the increased activities of phosphatase and urease hydrolyzed phosphate and nitrogen-containing organic matter into more free ions [12,38], and these ions were absorbed and utilized by plants, promoting plant growth and increasing the amount of root exudates, forming positive feedback. In contrast, the lower enzyme activities of $C_B$ treatment might be due to the dehydration condensation of the functional groups such as –COOH and –OH on the particle surface with the amino group (–NH$_2$) of the enzymes. Thus, the structures of functional groups or active sites of the enzymes were changed, affecting the interaction between the enzymes and the substrates and resulting in reduced enzyme activities [40]. The hydrophobic surface and porous structure of biochar can also adsorb and fix soil enzymes, causing conformational changes in enzyme active sites and reduced enzyme activity [7,40]. Moreover, the correlation analysis showed that soil enzyme activity was positively correlated with soil fertility, and the higher soil fertility under $C_C$ treatment was also a significant factor in improving the activities of soil enzymes.

### 4.4. Effects of Inorganic Fertilizer Level and Coffee Husk Returning Methods on Photosynthetic Characteristics of Coffee Trees

Photosynthesis is a critical process for plants to fix carbon dioxide and produce organic matter. Previous studies have shown that several factors, such as soil water supply, fertilizer amount, and light intensity, are significantly affecting crop photosynthetic efficiency. Our study also revealed a pattern of first increase and then decrease for net photosynthetic rate, transpiration rate, leaf water use efficiency, and stomatal conductance with the increase of

fertilization application. This pattern could primarily be attributed to the fact that proper fertilization supplemented soil nitrogen, increased chlorophyll content and chloroplast activity in crop leaves, and promoted respiratory electron transfer and stomatal opening. These factors are beneficial to photosynthetic pigments because they convert captured light energy into chemical energy at a higher rate and efficiency. Consequently, it improved photosynthetic performance and prolonged the green functional period of leaves, leading to improved crop photosynthetic efficiency [41]. Whereas excessive fertilization resulted in a decreased photosynthetic rate, this might be due to an increase in the osmotic potential of the soil solution. This increase reduced the water potential gradient between the soil and root system, leading to a decrease in stomatal conductance and resistance to water absorption by root cells. Therefore, it induced water and nutrient stress, weakening the photosynthesis performance [42].

The net photosynthetic rate, transpiration rate, and stomatal conductance of $C_C$ were significantly better than those of $C_B$ and $C_A$. This might be caused by the formation of a large amount of humic acid under the action of soil microorganisms after the compost of Cc was returned to the field. Under certain conditions, humic acid can alter the surrounding soil environment by stimulating the $H^+$-ATPase activity in plant roots and improving the plant's absorption of nutrient elements such as Na, K, and P, thereby promoting the increase of plant photosynthetic rate and stomatal conductance [43]. While $C_B$ changed the physical and chemical interface of the soil, root system, and biochar, it also affected the growth and distribution of the root system, reduced the survival rate of the root system, and directly inhibited the absorption of nutrients, resulting in reduced physiological activity of the plant [44,45]. Although other studies have shown that biochar has a higher effect on plant photosynthesis than compost [11,16], different from our research results. It might be due to the differences in soil properties. In addition, the enriching effect of biochar easily led to an increase in the concentration of harmful substances in the roots of plants and resulted in decreased activity of metalloproteinases in the root system, inhibiting the growth of crops [46]. The lower intercellular $CO_2$ concentration and leaf water use efficiency of $C_C$ might be because of the higher leaf photosynthetic rate. When the utilization of $CO_2$ is sufficient, the intercellular $CO_2$ concentration will decrease. At the same time, to ensure a sufficient supply of $CO_2$ in photosynthesis, the guard cells will absorb water and swell to keep the stomata open, thus causing the water use efficiency of leaves to decrease. Leaf water use efficiency was negatively correlated with soil fertility, microbial population, and enzyme activity, which might be because its own changes were directly affected by various environmental factors, such as soil moisture content.

### 4.5. Limitations and Future Perspectives

The research on agricultural and forestry waste mainly focuses on the return of straw compost to the field and the preparation of biochar, among which there have been relatively systematic achievements in the application of organic–inorganic fertilizers and soil amendments. However, in terms of the utilization of coffee waste, it mainly focuses on composting of coffee grounds, biochar, and deep processing of coffee husk [47,48]. Nevertheless, there is a lack of research on the soil environment and plant physiological status after coffee husk returning methods to the field with different treatment methods. By investigating the effects of different methods of combining coffee husks with inorganic fertilizer on soil properties and coffee growth characteristics, we addressed the crucial issue of improving soil fertility and crop growth. This study could contribute to developing appropriate coffee husk utilization methods and sustainable cultivation practices that are both efficient and environmentally friendly.

This experiment was mainly carried out in a greenhouse, and the simulation of the natural environment was not sufficient, so field experiments should be carried out in the next stage to obtain data closer to the real growth environment. In addition, the growth of young coffee trees is a complex process, and there are differences in the nutritional requirements of plants at different growth stages and seasons. In the next stage, detailed

research should be conducted to determine the optimal fertilization amount for different growth stages. In future research, it is essential to conduct an economic analysis of the proposed coffee husk return method to assess its feasibility in real-world agricultural contexts. Such economic analysis is crucial for practical applications and can provide valuable insights into the economic viability of the proposed method.

### 5. Conclusions

Medium fertilization levels combined with coffee husk compost ($F_M C_C$) not only met the needs of young coffee trees for nutrient amounts of elements but also demonstrated a remarkable improvement in soil organic matter, organic carbon, and total nitrogen, as well as an increase in the activities of urease, phosphatase, and catalase and the abundance of soil microorganisms. Furthermore, $F_M C_C$ significantly enhanced the net photosynthetic rate, transpiration rate, and stomatal conductance of Arabica coffee plants. Correlation analysis revealed a positive relationship between soil fertility and the photosynthetic index. Weight-TOPSIS $F_M C_C$ provided the highest comprehensive benefit. Medium fertilization levels combined with coffee husk compost were recommended as the optimal coupling methods for improving the soil environment in the root zone and enhancing the photosynthetic characteristics of Arabica coffee plants. These findings provide useful guidance for fertilization practices in Arabica coffee cultivation and offer insights into the returning of coffee husk and its comprehensive utilization, promoting the sustainable development of the coffee industry.

**Author Contributions:** Data curation, methodology, writing—original draft preparation, Z.J. and Y.L.; data collection and analysis, W.S.; writing—review and editing, funding acquisition, project administration, X.L.; Formal analysis, H.W., J.L., J.G., N.L. and Q.Y. All authors have read and agreed to the published version of the manuscript.

**Funding:** This study was supported by the National Natural Science Foundation of China (51979133) and Yunnan Fundamental Research Projects (grant No. 202301AS070030).

**Data Availability Statement:** Data from this work are available upon request.

**Acknowledgments:** The authors express their gratitude to the editor and the reviewers for their constructive comments

**Conflicts of Interest:** The authors declare no conflict of interest.

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
