# Peer review of "Combined Application of Coffee Husk Compost and Inorganic Fertilizer to Improve the Soil Ecological Environment and Photosynthetic Characteristics of Arabica Coffee"

_agronomy, doi:10.3390/agronomy13051212_

Round 1

Reviewer 1 Report

Journal “Agronomy” -2335050 manuscript examines the Combined application of coffee husk compost and inorganic fertilizer to improve the soil ecological environment and photosynthetic characteristics of Arabica coffee. This manuscript describes new and carefully confirmed findings, and experimental procedures which are given in sufficient details. The length of a full paper satisfies the minimum required to describe and interpret the work clearly. The manuscript is very well written, with relevant information that adequately addresses the given topic.

The style and language are largely in keeping with academic standards, providing clearly expressed information in a logical order and easy to follow.

Nevertheless, in the Discussion, please give a frank account of the strengths and weaknesses of the article. Include specific, detailed comments regarding the originality, scientific quality, relevance to the field of this journal, and presentation. Nevertheless, in the section Results, check and make it even tables and figures, and the adequacy of the references. Example: in the Introduction, Line 83: Cite certain studies about compost e.g (Filipović et al. 2023 " Productivity and flower quality of different pot marigold (Calendula officinalis L.) varieties on the compost produced from medicinal plant waste "., Filipović & Ugrenović, 2013 " The Composting Of Plant Residues Originating From The Production Of Medicinal Plants ". Studies should be in line with the circular economy.

Reviewer 2 Report

my comments:

please provide the pH of the soil after the end of the experiment

scale in all figs is not very detailed

Reviewer 3 Report

I think the experimental design described is not the correct one.

My suggestion for the experimental design is this: The experiment was carried out in a randomized block design, with three replications, in a subplots scheme over time, with the plots being composed of a 3x3 factorial (3 levels of inorganic fertilizer and 3 forms of coffee husk application). In the subplot, 4 evaluation times were used (insert the months of the evaluation).

I think the comparison used by Duncan's test is not the most correct test to be used, such as it was used. Due to the change in the experimental design, the statistical analyses must be changed.

My suggestion is these: Comparisons of means of coffee husk treatments (Ca, Cb, Cc) (qualitative treatments) should be carried out within each level of inorganic fertilizer application (Fl, Fm, Fh) (quantitative treatments) and linear regression models of fertilizer levels for each coffee husk use treatment (ca, cb, cc). For statistical analyses, analyses of variance were performed, with averages compared by Duncan's test (p< 0.05), regression models tested by F-test, and parameters tested by t-test (p<0.05).

Also, I suggest doing a principal component analysis to see the dispersion of the interaction between coffee husk treatments and doses of inorganic fertilizers.

Round 2

Reviewer 3 Report

A part of my suggestions was inserted into the text.

However, the most significant of them, referring to the methodological experimental design was not considered.

Despite the data collected and the discussion made being of high scientific value, I believe readers with knowledge of statistical methods will find the fault committed in the experimental design and in the respective analyzes of quantitative data and qualitative data that were analyzed all as quantitative data.

The manuscript has very high scientific value.

No file has been added for me here.

Best Regards